# Management of *Streptococcus mutans*-*Candida* spp. Oral Biofilms’ Infections: Paving the Way for Effective Clinical Interventions

**DOI:** 10.3390/jcm9020517

**Published:** 2020-02-14

**Authors:** Bahare Salehi, Dorota Kregiel, Gail Mahady, Javad Sharifi-Rad, Natália Martins, Célia F. Rodrigues

**Affiliations:** 1Student Research Committee, School of Medicine, Bam University of Medical Sciences, Bam 44340847, Iran; bahar.salehi007@gmail.com; 2Department of Environmental Biotechnology, Lodz University of Technology, 90-924 Lodz, Wolczanska 171/173, Poland; dorota.kregiel@p.lodz.pl; 3Department of Pharmacy Practice, Clinical Pharmacognosy Laboratories, University of Illinois at Chicago, Chicago, IL 60612, USA; mahady@uic.edu; 4Phytochemistry Research Center, Shahid Beheshti University of Medical Sciences, Tehran 1991953381, Iran; 5Department of Chemistry, Richardson College for the Environmental Science Complex, The University of Winnipeg, 599 Portage Avenue, Winnipeg, MB R3B 2G3, Canada; 6Faculty of Medicine, University of Porto, Alameda Prof. Hernâni Monteiro, Porto 4200-319, Portugal; 7Institute for Research and Innovation in Health (i3S), University of Porto, Porto 4200-135, Portugal; 8LEPABE—Laboratory for Process Engineering, Environment, Biotechnology and Energy, Faculty of Engineering, University of Porto, Rua Dr. Roberto Frias, Porto 4200-465, Portugal

**Keywords:** oral biofilm, infection control, *Streptococcus mutans*, *Candida* spp., natural compounds, antimicrobial resistance

## Abstract

Oral diseases are considered the most common noncommunicable diseases and are related to serious local and systemic disorders. Oral pathogens can grow and spread in the oral mucosae and frequently in biomaterials (e.g., dentures or prostheses) under polymicrobial biofilms, leading to several disorders such as dental caries and periodontal disease. Biofilms harbor a complex array of interacting microbes, increasingly unapproachable to antimicrobials and with dynamic processes key to disease pathogenicity, which partially explain the gradual loss of response towards conventional therapeutic regimens. New drugs (synthesized and natural) and other therapies that have revealed promising results for the treatment or control of these mixed biofilms are presented and discussed here. A structured search of bibliographic databases was applied to include recent research. There are several promising new approaches in the treatment of *Candida* spp.–*Streptococcus mutans* oral mixed biofilms that could be clinically applied in the near future. These findings confirm the importance of developing effective therapies for oral *Candida*–bacterial infections.

## 1. Introduction

According to the World Health Organization (WHO), oral diseases are the most common noncommunicable diseases, causing discomfort, pain, disfigurement, and death [1,2]. Dental caries, the most prevalent condition, derives from microbial biofilms (plaque) formed on the tooth surface [1,2]. Presently, it is estimated that 2.4 billion people have caries of permanent teeth and 486 million children have caries of primary teeth [3]. Similarly, periodontal disease (which affects tissues that both surround and support the teeth), and dental caries are both related to bacterial/fungal infections and are significant causes of deciduous tooth decay in over 560 million children, involving hundreds of billions of dollars of expenses per year [1,4]. In low-income populations, the majority of dental caries are left untreated, and affected teeth are most often extracted due to pain and discomfort [1]. The pain and inflammation of severe dental caries can impair eating and sleeping, as well as the overall quality of life. Abscesses can occur and may result in pain and chronic systemic infection and diseases [1]. If not treated, these disorders can also lead to chronic diseases and serious systemic infections (e.g., Alzheimer’s disease, cardiovascular disorders, oral cancer) [3,5].

Oral pathogens easily grow and propagate in the oral cavity, leading to the formation of dental plaque on both soft and hard tissue [6]. Dental plaque is formed by salivary molecules, proteins, bacterial/fungal debris, and sialic acid, which is then colonized by primary colonizers, including *Streptococcus sanguis* and *Actinomyces viscosus*. Their colonization is impacted by various food and environmental factors, including pH, carbon sources, and osmolarity [7,8,9]. Then, other bacterial species such as *Streptococcus mutans* adhere to the primary colonizers, and the tooth surface develops into a bacterial biofilm, known as dental plaque. Beyond dental plaque is dental calculus, which is a complex combination of dental plaque, salvia, and gingival crevicular fluid. The growth and invasiveness of oral pathogens are both regulated by an equilibrium between dental plaque bacteria and the innate immune system. The build-up of calcified dental plaque that extends into the subgingival layer can trigger inflammation due to perturbations of the immune system [7,8]. The calcified dental plaque can trap many biomolecules, viruses, and other bacteria and fungi, and alterations in the oral microbiome can lead to a wide range of diseases over a lifetime [8]. It has been suggested that the human oral microbiome is made up of over 2000 taxa of bacteria and fungi, including a wide range of opportunistic pathogens involved in cardiac, periodontal, respiratory, and other diseases, including cardiovascular and respiratory diseases, diabetes, and osteoporosis [8]. This review focuses on reports published in the last five years, related to one of the most important oral co-infections: *Streptococcus mutans*–*Candida* spp. biofilms.

### Streptococcus mutans and Candida *spp.*: Relevance in Oral Biofilms

The colonization of teeth by cariogenic bacteria is one of the most important risk factors in the development of dental diseases, with *S. mutans* being the primary species associated with the early dental caries process [10]. *Streptococcus mutans* is a Gram-positive, facultative anaerobic bacteria, and the primary etiological agent of dental plaques and dental caries [11,12]. This species is closely related to the streptococci group that inhabits the mouth, pharynx, and intestine and is well adapted to form biofilms due to its ability to form amyloids, which are very prevalent in natural biofilms [12]. Also, it colonizes the dental surface, causing damage to the hard tooth structure in the presence of fermentable carbohydrates, since it is a recognized acid-producing bacterium (e.g., sucrose and fructose). Moreover, *S. mutans* can adhere to the enamel salivary pellicle and other plaque bacteria, where they produce acidic metabolites, build up glycogen stores, and synthesize extracellular polysaccharides (EPS), glucans, and fructans from dietary sugars, leading to increased dental caries. In fact, there are several acidogenic and aciduric species in dental plaque associated with dental caries development, but *S. mutans* is the primary producer of EPS, which makes these biofilms difficult to control [11].

Along with the thousands of *Streptococcus* spp., over 100 fungal phylotypes also colonize the oral cavity, including many *Candida* spp. [13]. They are often found in the oral cavity of healthy individuals, with *Candida albicans* being the most predominant species (~60%–70%), followed by *Candida tropicalis* and *Candida glabrata* [13,14]. Usually, *Candida* spp. are commensal; however, in specific situations, these fungi may become parasitic, causing oral candidiasis particularly in immunocompromised patients, such as those with HIV/AIDS [13,15,16]. These fungal species are present in coating surfaces, dentine, as well as in the cementum surface. Interestingly, *C. albicans* grows in enamel cracks, grooves and flows into the crevices of cavities, as well as deeply penetrating open dentinal tubules [13]. Several *Candida* spp. have been isolated from caries as well as from root and dentinal caries in both children and adults, with prevalence ranging from 66% to 97% in pediatric populations and 31% to 56% in adult populations [15]. Thus, *C. albicans* can contribute to dentine and root caries in both children and adults.

Remarkably, *Candida* spp. cannot effectively form plaque biofilms directly or bind to *S. mutans* unless sugar is present. In a 2017 study, it was found that an enzyme known as glucosyltransferase B (GftB, which promotes firm cell clustering, and increased cohesion of plaque), secreted by *S. mutans*, uses sugar from the diet to manufacture glue-like polymers called glucans. *Candida* spp. increases glucan formation, causing the formation of a sticky biofilm that allows the yeast to adhere to teeth and bind to *S. mutans* [17]. These researchers further showed that the outer portion of the *Candida* spp. cell wall, composed of molecules called mannans, was involved in binding GftB, and that mutant *Candida* strains that lacked the mannan components had impaired binding to *S. mutans* GftB and a reduced biofilm load [17]. They further tested this hypothesis on biofilm formation in a rodent model of early childhood caries. Rodents infected with both *S. mutans* and either the wild type or defective mutant *Candida* strains showed that animals infected with wild-type *Candida* spp. had abundant biofilm formation, while those infected with the mutant *Candida* strains had reduced biofilm formation by up to ~5-fold [17].

Glucosyltransferases (Gtfs)-derived EPS have been revealed to be a key mediator of co-species biofilm development and their co-existence with *C. albicans* induces the expression of virulence genes in *S. mutans.* In fact, water-soluble glucan synthesis is regulated by the *dexA* gene and affects biofilm aggregation and cariogenic pathogenicity in *S. mutans* [18]. The co-cultivation altered *S. mutans* signal transduction and the transcription of genes (*comC* and *ciaRH*) associated with fitness and virulence [19]. Arzmi and co-workers [20] suggested that polymicrobial biofilms differentially modulate oral microorganisms’ phenotypes. Yang and co-workers [21] found that antigen I/II of *S. mutans* is important for *Candida* spp. incorporation into the biofilm and is also required for increased acid production. *S. mutans* not only modulates biofilm formation, but also attenuates *C. albicans* virulence [22,23]. Additionally, the possibility of a *quorum sensing* system stimulation of *S. mutans* by *C. albicans* was also demonstrated, consequently changing its virulence properties [24].

## 2. Research Methodology

A structured search of bibliographic databases (PubMed Central, Elsevier’s ScienceDirect, SCOPUS, and Springer’s SpringerLink) was undertaken. The keywords used were “streptococcus” + “mutans” + “candida” + “biofilms” + “resistance”.

## 3. Compounds with Activity against Oral Infections

Oral microbial communities are some of the most complex microbial floras in the human body, consisting of thousands of bacterial and hundreds of fungal species [25]. Culture-independent molecular methods, such as proteomics and 16S rRNA sequencing, have demonstrated that *S. mutans* was the dominant species, with elevated levels of other streptococci including *Streptococcus sanguinis*, *Streptococcus mitis*, and *Streptococcus salivarius*. *Candida* spp. combined with *Streptococcus* spp. usually increases the virulence in invasive candidiasis, early childhood caries, or peri-implantitis [13,26,27]. The interactions between the various species in these mixed biofilms may be synergistic, in that the presence of one microorganism generates a niche for other pathogenic microorganisms, which serves to facilitate organisms’ retention. Thus, *Streptococcus* spp. and *Candida* spp. create special synergistic consortia on solid oral surfaces [13,26].

*Streptococcus mutans* strains are considered the most cariogenic bacteria; however, *Candida* spp. can increase their cariogenicity. Bacterial and fungal cells are able to produce glucan as an extracellular polysaccharide that aids cariogenic biofilm formation [28]. Bacteria also convert dietary sucrose and free glucose or fructose into a diverse range of soluble and, particularly, insoluble extracellular polysaccharides (EPS) (e.g., water-insoluble glucans) through exoenzymes, such as Gtfs, and EPS, to build blocks of cariogenic biofilms. These structures promote the tooth surface colonization by *S. mutans* and additional microorganisms into dental plaque (e.g., *Porphyromonas gingivalis*, streptococci, *Fusobacterium* spp., *Prevotella* spp.), while forming the scaffold core or matrix of the biofilm. In addition, the EPS matrix creates a special diffusion-limiting barrier, facilitating acidic microenvironments creation at the biofilm–tooth interface, which is critical for adjacent tooth enamel dissolution [29]. Furthermore, *S. mutans* displays many other virulence attributes, including its ability to produce acid and to tolerate an acidic environment.

Among *Candida* spp., *C. albicans* is the most prevalent on dental tissues, reaching 60%–70%, followed by *C. tropicalis* and *C. glabrata* [30]. As indicated earlier, these yeasts are usually commensal, but in some situations they can become parasitic, causing oral candidiasis [30,31]. The virulence attributes of *C. albicans* are the acidogenicity and aciduric nature, along with the ability to develop profuse biofilms, to ferment and assimilate dietary sugars, and to produce collagenolytic proteinases [15]. Also, *C. albicans* modulates the pH in dual-species biofilms to values above the critical pH where enamel dissolves. Although the species present low cariogenicity, biofilms formed by mixed populations of *C. albicans* and *S. mutans* are more voluminous [32]. The presence of *Candida* spp. enhances *S. mutans* growth, fitness, and accumulation within biofilms. It is documented that *C. albicans* growth stimulates *S. mutans* development via biofilm-derived metabolites [33]. Also, *C. albicans* mannans mediate *S. mutans* exoenzyme GtfB binding to modulate cross-kingdom biofilm development [17]. *Candida* spp.-derived β-1,3-glucans contribute to the EPS matrix structure, while fungal mannan and β-glucan provide sites for GtfB binding and activity; thus, the coexistence of *S. mutans* with *C. albicans* can cause dental caries progression or disease recurrence in the future [34]. It was also suggested that *S. mutans* co-cultivation with *C. albicans* influences carbohydrate utilization by bacterial cells [35,36], and the analysis of metabolites confirmed the increase in carbohydrate metabolism, with amounts of formate elevated by co-cultured biofilms [37]. Nonetheless, biofilm biomass and metabolic activity were both strain- and growth medium-dependent [37]. Liu and co-workers showed that nicotine promotes *S. mutans* attachment. The enhancement of the synergistic relationship may contribute to caries development in smokers [38,39]. On the other hand, there are contradictory reports in this matter. It was proposed that, by definition, *C. albicans* is not a cariogenic microorganism; it could prevent caries by actively increasing pH and preventing mineral loss [40]. Nonetheless, a study published in 2016 concluded that children with severe early childhood caries were likely infected by their mothers, as the mothers of these children were highly infected with *C. albicans* [41]. In fact, genetic testing of *Candida* strains from children and their mothers showed that most strains were genetically related [41].

Progress in understanding the etiology, epidemiology, and microbiology of periodontal pocket flora has called for new antimicrobial therapeutic schemes for oral diseases [42]. Distinctive means and compounds may be used to prevent oral infections. Biomolecules produced by *S. mutans*, lactoferrin, and probiotics, applying chemicals and photodynamic therapy, can support the management of oral candidiasis [43].

### 3.1. Chemical Compounds with Activity against Streptococcus mutans and Candida *spp.* Biofilms

Recently, a wide range of antimicrobial agents and methodologies have been reported to suppress the growth of dental *Streptococcus mutans–Candida* spp. biofilms (Table 1).

Photodynamic antimicrobial therapy (aPDT) is based on a photoactive dye—a chemical photosensitizer that binds to the target cell and is activated by a specific light wavelength. During this process, oxygen species such as singlet oxygen and free radicals are formed, exerting toxicity towards the cell [30]. The influence of pre-irradiation time employed in antimicrobial photodynamic therapy with a diode laser was recently checked [79]. It was found no effect due to *C. albicans* biofilm therapy; however, a pre-irradiation time of 1 min was effective against a microbial load of *S. mutans*. The effect of aPDT using chloroaluminium phthalocyanine in a cationic nanoemulsion was evaluated against multispecies biofilms of *C. albicans, C. glabrata*, and *S. mutans*. The technique led to photoinactivation of the biofilm and reduced colony count and metabolic activity [44]. Similar effects were obtained using hypericin–glucamine [45], Rose Bengal in α-cyclodextrin [46], curcumin [44,47,48,49,50], methylene blue [52], toluidine blue O [51], erythrosine with green light [55], or Photodithazine^®^ obtained from the cyanobacterium *Spirulina platensis* [53,54]. Another study conducted by Soria-Lozano et al. [80], using Rose Bengal, methylene blue, and curcumin with white light, showed positive effects against the *S. mutans* and *S. sanguis* strains. Although methylene blue and Rose Bengal were the most efficient, *C. albicans* was the most resistant to all photosensitizers, and curcumin, in this case, was ineffective. Finally, aPDT embodies an important treatment as the photodynamic inactivation seems to be promising for biofilm-associated *S. mutans* and *Candida* spp. biofilm infection management, since there is no microbial resistance observed.

Chlorhexidine is the most commonly studied active agent [81,82]. It is widely used for preventing dental plaque or treating mouth yeast infections. Additionally, over the years, the number of studies on antibiofilm strategies using this compound or other chemicals has increased [83]. Chlorhexidine at low concentrations, but with the addition of cis-2-decenoic acid, was able to disperse single-species biofilms formed by *S. mutans* and *C. albicans*, as well as bacterial–fungal dual-species consortia [56]. Also, chlorhexidine gluconate with tyrosol was revealed to be effective against single and mixed-species oral biofilms [57,58]. A solution containing 2.5% sodium hypochlorite, 2% chlorhexidine, and ozonated water inhibited biofilms of *S. mutans* and *C. albicans* in mesiobuccal root canals after irrigation [25]. A chlorhexidine carrier nanosystem based on iron oxide magnetic nanoparticles and chitosan was synthesized, and its antimicrobial effect on mono- and dual-species biofilms of *C. albicans* and *S. mutans* was evaluated. The results confirmed the nanosystem potential as a preventive or therapeutic agent to fight biofilm-associated oral diseases [60]. The advantage of chlorhexidine is that its resistance is not as common as other chemicals’. Importantly, cases are being reported more often [84]. Hence, in order to reduce the number of cases, it is important to restrict its applications to indications with a strong patient benefit and to remove it from uses that are without any advantage or have doubtful value.

The antimicrobial activity of chitosan, silver nanoparticles, and ozonated olive oil was evaluated against endodontic pathogens, including *S. mutans* and *C. albicans*. This combination was characterized as novel, safe, and having the potential to eradicate mature mixed-species biofilms [61]. Chitosan and carboxymethyl chitosan were also active against *Candida* spp. and *Streptococcus* spp. biofilms [62,63,64,65]. Also, Kivanç and co-workers [72] investigated the antimicrobial and antibiofilm activities of hexagonal boron nitride nanoparticles against *S. mutans*, *Staphylococcus pasteuri*, *S. mutans -Candida* spp. Their results showed that, at an appropriate concentration (0.1 mg/mL), these nanoparticles could be considered a safe potential oral care product.

Similarly, EPS inhibitors may enhance antibiofilm activity [85,86]. A combination of the antifungal fluconazole with povidone iodine showed to completely inhibit *C. albicans* or mixed biofilm formation [76]. It was found that the inclusion of iodine derivative enhanced fluconazole efficacy by inhibiting α-glucan synthesis in *S. mutans*, which participates in protective bacterial EPS formation [76]. Moreover, thiazolidinediones have been found to act as effective *quorum sensing* quenchers, capable of preventing the mixed biofilm formed by *Candida* spp. and bacterial strains. These compounds can penetrate into deeper layers of the mixed biofilm, thereby increasing the antimicrobial activity. A small molecule, thiazolidinedione-8, has been revealed to be able to impair biofilm formation of various microbial pathogens. These compounds may disturb the symbiotic balance between *C. albicans* and *S. mutans* in a dual-species biofilm [77]. The inhibitory effects of lactams in mixed oral biofilms, including *S. mutans* and *C. glabrata*, have been assessed [78]. γ-Alkylidene-γ-lactams solubilized in 3.5% dimethyl sulfoxide led to a marked reduction in biofilm biomass. Furthermore, the total protein content and the quantity of EPS declined significantly. Hence, these compounds show important antibiomass activity, which can be important for promoting the diffusion of a second drug into oral biofilms, for its total eradication.

Several materials containing silver nanoparticles, alone or with polyphosphates, were evaluated as antimicrobials against *C. albicans* and *S. mutans*. These composites demonstrated significant antimicrobial activity, especially against *S. mutans*, which might make them a possible alternative for new dental materials [66,67]. Silver nanoparticles, combined with calcium glycerophosphate as well as nanostructured silver vanadate in dental acrylic resins, have also been shown to have antimicrobial and antibiofilm activities against dental prostheses-associated microorganisms [73,74]. Silver has found several uses since its toxicity toward human cells is considerably lower than toward bacteria; thus, this application is promising.

Base resins have also shown promising bioactive responses against *C. albicans* and *S. mutans.* Dimethylaminododecyl methacrylate-modified denture base resin proved to be a favorable therapeutic system against problems triggered by denture base microbes (e.g., denture stomatitis) [68]. Likewise, poly-(2-tert-butilaminoethyl) methacrylate decreased *S. mutans*’ adhesion to the material surface, but did not exhibit an antimicrobial effect against *C. albicans* [69]. Furthermore, a novel fluoride-releasing copolymer composed of methyl methacrylate, 2-hydroxyethyl methacrylate with polymethyl methacrylate was developed by incorporating sodium fluoride. This copolymer inhibited acidogenic mixed-species biofilms, showing potential to control these diseases by limiting biofilm growth [59]. Finally, a modified pH-responsive cationic poly (ethylene glycol)-block-poly (2-(((2-aminoethyl) carbamoyl) oxy) ethyl methacrylate has been revealed to be a promising agent for dental caries therapy and provided guidelines for drug delivery system design in other acidic pathologic systems [75].

Regarding synthetic surfactants, several compounds have also shown great potential. Cetylpyridinium chloride and cetyltrimethylammonium bromide with plant terpinen-4-ol revealed antimicrobial activity against *S. mutans* and *C. albicans* [70]. Nanoparticles of amphiphilic silanes with Chlorin e6 exhibited strong antibiofilm activity against periodontitis-related pathogens belonging to the *Streptococcus* genus [71]. These compounds have good prospects in antimicrobial applications to inhibit both oral disease occurrence and progression, namely periodontitis, one of the most relevant oral diseases.

### 3.2. Natural Compounds with Bioactivity Against Streptococcus mutans and Candida *spp.* Biofilms

Due to the incidence of oral disease, increased resistance by bacteria and fungi to antimicrobials, and the adverse effects of some drugs currently used in dentistry (and general medicine), there is a great need for alternative prophylaxis and treatment options that are not only safer but also cost-effective. While several antimicrobial agents are commercially available, these chemicals can alter the oral microbiota and have undesirable side effects (e.g., nausea, diarrhea, vomiting, tooth staining). Henceforth, traditional medicine and the search for alternative products are still important. In fact, natural products offer an assortment of chemical structures and possess an extensive variety of biological properties; thus, they are a source for new pharmaceuticals. Bioactive secondary metabolites have been revealed to be useful as new antimicrobial and antibiofilm drugs, such as numerous furanones, alkaloids, and flavonoids, from many plants and marine organisms [87]. Table 2 presents natural compounds and plant extracts with antimicrobial potential, particularly focused on anti-*Streptococcus* spp. anti-*Candida* spp. activities.

Plant-derived compounds are a good source of therapeutic agents and inhibitors of dental caries, periodontal diseases, and candidiasis. For example, α-mangostin or lawsone methyl ether showed antimicrobial, antibiofilm, and anti-inflammatory activities. Indeed, an oral spray containing these natural chemicals was effective against common oral pathogens [98]. Lupinifolin from *Albizia myriophylla* wood exhibits good anti-*S. mutans* activity by damaging bacterial membranes, resulting in cell leakage [99]. Essential oils and bioactive fractions from *Aloysia gratissima*, *Baccharis dracunculifolia*, *Coriandrum sativum*, *Cyperus articulatus*, and *Lippia sidoides* were also evaluated as antimicrobials against *S. mutans* and *Candida* spp. A significant reduction in extracellular polysaccharides and bacteria was observed for *A. gratissima* and *L. sidoides*, indicating that these fractions disrupted biofilm integrity. Plus, *C. sativum* oils drastically affected *C. albicans* viability [100], and might be considered as alternative anticaries agents. In turn, quercetin and kaempferol or farnesol with myricetin showed favorable properties in terms of controlling some virulence factors of *S. mutans* and *C. albicans* biofilms [101,102,103,104]. It was suggested that tyrosol decreased the metabolic activity and number of viable cells in single and mixed-species biofilms [105]. Correspondingly, β-caryophyllene—a bicyclic sesquiterpene of numerous essential oils—may inhibit cariogenic biofilms and may be a candidate agent for the prevention of dental caries [28]. These results highlight the promising antimicrobial activity of plants for the treatment of dental caries and oral candidiasis.

Propolis, rich in flavonoids, has a long history of use as a natural treatment for a host of health problems. It is also used as an ingredient in certain medicinal products applied directly to the skin, or in mouthwash and toothpaste. Propolis is under preliminary research for the potential development of new drugs associated with the control of *C. albicans* and immunomodulatory effects. Propolis and miswak (*Salvadora persica* tree), used in toothpaste, dental varnishes, and mouthwash, led to a significant reduction in the colony-forming units of oral biofilms [106,107].

Antimicrobial peptides have also been widely tested for controlling bacterial biofilms. Shang and colleagues [108] tested the efficacy of peptides from *Rana chensinensis* skin secretions in preventing biofilm formation by cariogenic and periodontic pathogens. Peptide L-K6, a temporin-1CEb analog, exhibited high antimicrobial activity against tested oral pathogens and was able to inhibit *S. mutans* biofilm formation. This peptide significantly reduced cell viability within oral biofilms. Its anti-inflammatory activity was correlated with L-K6 binding to LPS and dissociating LPS aggregates. Likewise, cyclic dipeptides (CDPs) are common metabolites widely biosynthesized by cyclodipeptide synthases or nonribosomal peptide synthetases by both prokaryotic and eukaryotic cells. Examples of CDPs that inhibit biofilm formation by *Streptococcus epidermidis* include cyclo-(l-Leucyl-l-Tyrosyl) isolated from mold *Penicillium* sp. and cyclo-(l-Leucyl-l-Prolyl) isolated from *Bacillus amyoliquefaciens.* The antibiofilm activity of 75 synthetic CDPs was assessed against oral pathogens, allowing for the identification of five novel CDPs that inhibit biofilm formation and adherence properties. Among them, five new active compounds were identified as preventing biofilm formation by *S. mutans* and *C. albicans* on the hydroxylapatite surface [109].

Similarly, the application of probiotics and antagonistic microorganisms for oral pathogens can bring about tangible benefits, namely boosting host immunity and disturbing the pathogen’s environment via competition for space and nutrients. Krzyściak et al. [35] evaluated the anticariogenic effects of *Lactobacillus salivarius* by reducing pathogenic species in biofilm models. This microorganism has demonstrated the ability to secrete intermediates capable of inhibiting the formation of cariogenic *S. mutans* and *C. albicans* biofilms. In other studies, single and mixed biofilms were inhibited by probiotic lactobacilli [110,111]. Therefore, they seem to be useful as an adjunctive therapeutic mode against oral *Candida* spp. infections. Another study reported that the *Candida sorbosivorans* SSE-24 strain was used to stimulate erythritol production at a level of 60 g/L. It was detected a significant inhibitory effect of erythritol on the growth and biofilm formation of *S. mutans* [112]. Interestingly, a novel phage (ɸAPCM01) with activity against *S. mutans* biofilms was recently isolated from human saliva [113], and the inhibition of *S. mutans* biofilms was also found by the use of liamocins from *Aureobasidium pullulans* [114]. Chemically, liamocins are mannitol oils specific to *Streptococcus* spp., having the potential to act as new inhibitors of oral streptococcal biofilms that should not affect normal oral microflora.

## 4. Conclusions and Future Prospects

Oral diseases continue to increase despite the best efforts of the medical and scientific communities. The most complicated pathologies derive from microbial biofilms (plaque), formed by a consortium of microorganisms, which are protected by a net of polymers (e.g., EPS, DNA). The biofilm matrix delays or blocks the antimicrobials’ diffusion, making treatment much more difficult or even unsuccessful. The oral *S. mutans*–*Candida* spp. mixed biofilm has been subject to various studies involving alternative therapeutics (e.g., aPDT), new chemical structures (natural and synthetic) with antimicrobial and antibiofilm activities, and nanotechnology, revealing different but promising antimicrobial properties. Nonetheless, more and deeper studies involving in vivo and clinical approaches are still needed.

## Figures and Tables

**Table 1 jcm-09-00517-t001:** Chemical compounds and their bioactivity against *Streptococcus* spp. and *Candida* spp.

Main Effect(s)	Compound	Targeted Species	Reference(s)
Pathogen toxicity	aPDT: chloroaluminium phthalocyanine (cationic nanoemulsion); hypericin-glucamine Rose Bengal in α-cyclodextrin; curcumin; methylene blue; toluidine blue O erythrosine with green light or Photodithazine^®^; rose Bengal, methylene blue and curcumin with white light	Mainly *C. albicans*, *C. glabrata*, *S. mutans*, and *S. sanguis*, but also other species	[44,45,46,47,48,49,50,51,52,53,54,55]
Antibiofilm	Chlorhexidine with low concentrations but with the addition of cis-2-decenoic acid	*S. mutans* and *C. albicans*, and bacterial–fungal dual-species consortia	[56]
	Chlorhexidine gluconate with tyrosol	Single and mixed-species oral biofilms	[57,58]
	Fluoride-releasing copolymer: methyl methacrylate, 2-hydroxyethyl methacrylate with polymethyl methacrylate (incorporating sodium fluoride)	Acidogenic mixed-species biofilms	[59]
Antimicrobial	2.5% sodium hypochlorite, 2% chlorhexidine, and ozonated water	Mono- and dual-species biofilms of *S. mutans* and *C. albicans*	[25]
	Chlorhexidine carrier nanosystem based on iron oxide magnetic nanoparticles and chitosan		[60]
	Chitosan, silver nanoparticles and ozonated olive oil	Several endodontic pathogens, including *S. mutans* and *C. albicans* mixed biofilms	[61]
	Chitosan and carboxymethyl chitosan	Biofilms of *Streptococcus* spp.and *Candida* spp.	[62,63,64,65]
	Materials containing silver nanoparticles alone or with polyphosphates	*C. albicans* and *S. mutans*	[66,67]
	Dimethylaminododecyl methacrylate modified denture base resin	Several microorganisms associated on the dental base	[68]
	Poly-(2-tert-butilaminoethyl) methacrylate	*S. mutans*	[69]
	Cetylpyridinium chloride and cetyltrimethylammonium bromide with plant terpinen-4-ol (Synthetic surfactants+)	*S. mutans* and *C. albicans*	[70]
	Nanoparticles of amphiphilic silanes with Chlorin e6	*Streptococcus* genus	[71]
Antimicrobial and antibiofilm	Hexagonal boron nitride nanoparticles	*S. mutans, Staphylococcus pasteuri*, *S. mutans-Candida* spp.	[72]
	Silver nanoparticles combined to calcium glycerophosphate or nanostructured silver vanadate (dental acrylic resins)	Several microorganisms associated with dental prostheses	[73,74]
	Modified pH-responsive cationic poly (ethylene glycol)-block-poly (2-(((2-aminoethyl) carbamoyl) oxy) ethyl methacrylate	Acidogenic mixed-species biofilms	[75]
Extracellular polysaccharides (EPS) inhibitors	Combination fluconazole with povidone iodine	*C. albicans* or mixed biofilm formation with *S. mutans*	[76]
	Thiazolidinediones, such as thiazolidinedione-8	Mixed biofilm formed by *Candida* spp. and bacterial strains (*S. mutans*)	[77]
	Lactams, such as γ-Alkylidene-γ-lactams solubilized in 3.5% dimethyl sulfoxide	*S. mutans* and *C. glabrata*	[78]

**Table 2 jcm-09-00517-t002:** Natural compounds/extracts and their in vitro bioactivity against *Streptococcus* spp. and *Candida* spp.

Year	In vitro Assays	Natural Compound/Extract	Effect	Reference(s)
2018	Antibacterial and antifungal bioactivity	*Acacia arabica* (extract)	Antibacterial source of anticariogenic agents	[88]
2018	Antibacterial and antifungal bioactivity	*Myracrodruon urundeuva* and *Qualea grandiflora* (hydroalcoholic extracts)	Activity against *S. mutans* biofilm	[89]
2018	Antibacterial and antifungal bioactivity	*Cissampelos torulosa*, *Spirostachys africana, Clematis brachiata, Englerophytum magalismonatanum* (extracts)	Activity against both *Streptococcus* spp. and *Candida* spp.	[90]
2017	Antibacterial and antifungal bioactivity Cytotoxicity and genotoxicity: murine macrophages (RAW 264.7), human gingival fibroblasts (FMM-1), human breast carcinoma cells (MCF-7), and cervical carcinoma cells (HeLa)	*Thymus vulgaris* and *Rosmarinus officinalis* (extracts)	Antimicrobial and anti-inflammatory effects against oral pathogens	[91,92]
2017	Antibacterial and antifungal bioactivity Antibacterial, antifungal, and antiadhesion in a tissue conditioner	*Azadirachta indica* (leaf extract)	Potential antimicrobial agent against both *S. mutans* and *C. albicans*	[93]
2017	Antibacterial and antifungal, antibiofilm and antioxidant bioactivity	*Camellia japonica* and *Thuja orientalis*	Significantly inhibited the microbial grow of oral pathogens	[94]
2016	Antibacterial and antifungal, antibiofilm bioactivity Cytotoxicity and anti-inflammatory effects: human oral epithelial cells	*Houttuynia cordata* (herbal tea)	Antibiofilm effects against *S. mutans* and *C. albicans*	[95]
2015	Antibacterial and antifungal bioactivity	*Ricinus communis* and sodium hypochlorite (cleanser solutions)	Effective in controlling denture biofilms	[96]
2014	Anti-adherent properties (antibiofilm)	*Schinus terebinthifolius* and *Croton urucurana* (methanol and acetate methanol extract fractions in hydroalcoholic and dimethylsulfoxide)	Antibiofilm activity against *S. mutans* and *C. albicans*	[97]

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
