# Peer review of "Management of Streptococcus mutans-Candida spp. Oral Biofilms’ Infections: Paving the Way for Effective Clinical Interventions"

_jcm, 2020, doi:10.3390/jcm9020517_

Round 1

Reviewer 1 Report

Generally, this manuscript is not well organized and has grammatical and syntax errors in some parts. I have two suggestions.

Since this manuscript tries to describe the control of Candida spp.-Streptococcus mutans oral biofilms, it would be better to introduce the pathogenic characteristics of Streptococcus mutans and Candida spp., and the relationship between them (relevance in oral biofilms) in the introduction part. It would be better to sort those compounds (chemically synthesized or natural products) based on their function (such as cell toxicity, EPS inhibitor, adherence inhibitor, anti-inflammatory activity, etc). It’s a little rough-and-tumble just by listing all the compounds.

Author Response

The authors thank the Reviewer for the important suggestions and recognition of the work done in this research. We tried to address all reviewer’s queries and hope to have clarified all points. An English review was done, as suggested.

Generally, this manuscript is not well organized and has grammatical and syntax errors in some parts. I have two suggestions.

Answer: The authors have revised the manuscript.

Since this manuscript tries to describe the control of Candida spp.-Streptococcus mutans oral biofilms, it would be better to introduce the pathogenic characteristics of Streptococcus mutans and Candida spp., and the relationship between them (relevance in oral biofilms) in the introduction part.

Answer: This information was added in this section, as suggested.

It would be better to sort those compounds (chemically synthesized or natural products) based on their function (such as cell toxicity, EPS inhibitor, adherence inhibitor, anti-inflammatory activity, etc). It’s a little rough-and-tumble just by listing all the compounds.

Answer: This change was done accordingly, as reviewer 2 also requested. In this sense, we created a table with the information reorganized.

Reviewer 2 Report

This review provides an interesting summary of the literature on compounds effective against oral S mutants and Candida biofilms. Overall it is well written and nicely presents potential future therapeutics. 

Specific comments:

Figure 1: It is unusual to see an image such as this captured by the authors in a review article, and does not add anything to the manuscript. In addition, methods to generate the image are lacking sufficient detail. I would suggest removing and/or replacing the figure with a diagram (e.g. cartoon) that represents the physical, biological and myco/bacteriological environment that S. mutans and C albicans biofilms occur in, to set the scene for the review.

Section 4.1: Authors present a nice table showing the natural compounds and their details (Table 1). It would be nice to see an additional table summarising what is described in section 4.1 pertaining to Chemical compounds as well. 

Minor comments:

Line 135-6 can the authors provide specific examples of extracellular polysaccharides produced by bacteria with citations to match?

Lines 139: additional microorganisms- please give examples of other microorganisms in dental plaque that are important.

Line 145: Provide citation for the % numbers used in Line 144-145. 

Line 168-169: The sentence preceding Reference 35 is, word for word, the title of reference 35 copy and pasted. Please adjust this sentence to make more meaningful statement of the work described in the article. 

Line 174: change attenuate to attenuates

Line 175: double spacing before 'the quorum sensing'

Line 186: insert 'of' between wavelength and light

Line 187: Replace 'producing toxicity to the cell' with exerting toxicity towards the cell'

Line 200: Check the use of the word 'Settling'

Line 202: change 'no microbial resistance' to 'yet to be microbial resistance'

Author Response

The authors thank the Reviewer for the important suggestions and recognition of the work done in this research. We tried to address all the reviewer’s queries and hope to have clarified all points.

This review provides an interesting summary of the literature on compounds effective against oral S mutants and Candida biofilms. Overall it is well written and nicely presents potential future therapeutics. 

Specific comments:

Figure 1: It is unusual to see an image such as this captured by the authors in a review article, and does not add anything to the manuscript. In addition, methods to generate the image are lacking sufficient detail. I would suggest removing and/or replacing the figure with a diagram (e.g. cartoon) that represents the physical, biological and myco/bacteriological environment that S. mutans and C albicans biofilms occur in, to set the scene for the review.

Answer: The figure was removed, as suggested by the reviewer.

Section 4.1: Authors present a nice table showing the natural compounds and their details (Table 1). It would be nice to see an additional table summarizing what is described in section 4.1 pertaining to Chemical compounds as well. 

Answer: This table was added in the revised manuscript.

Minor comments:

Line 135-6 can the authors provide specific examples of extracellular polysaccharides produced by bacteria with citations to match?

Answer: Examples and references were indicated.

Lines 139: additional microorganisms- please give examples of other microorganisms in dental plaque that are important.

Answer: As asked, this information was added in the revised manuscript.

Line 145: Provide citation for the % numbers used in Line 144-145. 

Answer: As asked, this information was added in the revised manuscript.

Line 168-169: The sentence preceding Reference 35 is, word for word, the title of reference 35 copy and pasted. Please adjust this sentence to make more meaningful statement of the work described in the article. 

Answer: This information was corrected and adjusted in the revised manuscript.

Line 174: change attenuate to attenuates

Answer: Corrected.

Line 175: double spacing before 'the quorum sensing'

Answer: Corrected.

Line 186: insert 'of' between wavelength and light

Answer: Corrected.

Line 187: Replace 'producing toxicity to the cell' with exerting toxicity towards the cell'

Answer: Replaced, as indicated.

Line 200: Check the use of the word 'Settling'

Answer: Corrected.

Line 202: change 'no microbial resistance' to 'yet to be microbial resistance'

Answer: Corrected.

Round 2

Reviewer 1 Report

The revised manuscript is better in writing and organization. It would be better to move 167-226 to the introduction since it mainly describes the relationship between S.mutans and Candida spp (The suggestion 1 in the previous comment). The description about characteristics of Streptococcus mutans and Candida spp. could be simplified since some of characteristics would be mentioned in relationship part.

Please move line 108-112 to the part of describing the characteristics of Candida spp..

Minor comment:

Line 25, please change “originating” to “leading to”.

Line 91, please change two “in” to “on”, should be “on the surface”.

Line 257, please change “is yet to be” to “remain no”.

Line 271, please remove “yet”.

Line 289, 317, 336, please change “revealed” to “been revealed”.

Line 326, please change “ones” to “oral diseases”.

Line 402, please keep “subject to”.

Author Response

The revised manuscript is better in writing and organization. It would be better to move 167-226 to the introduction since it mainly describes the relationship between S.mutans and Candida spp (The suggestion 1 in the previous comment). The description about characteristics of Streptococcus mutans and Candida spp. could be simplified since some of characteristics would be mentioned in relationship part.

Answer: The requested change was done

Please move line 108-112 to the part of describing the characteristics of Candida spp..

Answer: The requested change was done

Minor comment:

Line 25, please change “originating” to “leading to”.

Line 91, please change two “in” to “on”, should be “on the surface”.

Line 257, please change “is yet to be” to “remain no”.

Line 271, please remove “yet”.

Line 289, 317, 336, please change “revealed” to “been revealed”.

Line 326, please change “ones” to “oral diseases”.

Line 402, please keep “subject to”.

Answer: All changes were done.

Reviewer 2 Report

Authors have correctly addressed my previous comments

Author Response

Thank you for the overall appreciation of our work.